# Acute exercise as active inference in chronic musculoskeletal pain, effects on gait kinematics and muscular activity in patients and healthy participants: a study protocol for a randomised controlled laboratory trial

Jens Westergren [iD],[1] Veronica Sjöberg [iD],[1] Linda Vixner [iD],[1] Roger G Nyberg [iD],[2] David Moulaee Conradsson [iD],[3,4] Andreas Monnier [iD],[1,3] Riccardo LoMartire [iD],[5] Paul Enthoven [iD],[6] Björn O Äng [iD] [1,3,5,7]

For numbered affiliations see end of article.

**Correspondence to**
Jens Westergren; jws@du.se

## ABSTRACT

**Introduction** Chronic musculoskeletal pain is a highly prevalent, complex and distressing condition that may negatively affect all domains of life. In view of an active inference framework, and resting on the concept of allostasis, human movement per se becomes a prerequisite for health and well-being while chronic pain becomes a sign of a system unable to attenuate an allostatic load. Previous studies on different subgroups of chronic pain conditions have demonstrated alterations in gait kinematics and muscle activity, indicating shared disturbances in the motor system from long-term allostatic load. We hypothesise that such alterations exist in heterogenous populations with chronic musculoskeletal pain, and that exposure to acute and controlled exercise may attenuate these alterations. Therefore, the main aim of this study is to investigate the acute effects of exercise on gait kinematics and activity of the back and neck muscles during diverse walking conditions in patients with chronic musculoskeletal pain compared with a reference sample consisting of healthy participants.

**Methods and analysis** This two-sample two-armed parallel randomised controlled laboratory trial will include 40 participants with chronic musculoskeletal pain (>3 months) and 40 healthy participants. Participants will be randomly allocated to either 30 min of aerobic exercise or rest. Primary outcomes are gait kinematics (walking speed, step frequency, stride length, lumbar rotation, gait stability) and muscular activity (spatial and temporal) of the back and neck during diverse walking conditions. Secondary outcomes are variability of gait kinematics and muscle activity and subjective pain ratings assessed regularly during the trial.

**Ethics and dissemination** The study has been approved by the Regional Ethics Review Board in Uppsala, Sweden (#2018/307). Findings will be disseminated via conference presentations, publications in peer-reviewed journals and engagement with patient support groups and clinicians.

**Trial registration number** NCT03882333.

## STRENGTHS AND LIMITATIONS OF THIS STUDY

⇒ To our knowledge, this will be the first randomised controlled laboratory trial investigating the acute effects of aerobic exercise, as active inference, on gait kinematics and muscular activity in a heterogenous group of patients with chronic musculoskeletal pain.

⇒ As long-term aerobic exercise is ultimately the result of regular sessions of aerobic exercise, studying effects induced by a single session of aerobic exercise will provide important insights into exercise and its role in the rehabilitation of chronic pain conditions.

⇒ The study will incorporate objective measures of gait kinematics and muscle activity from continuous overground walking by using modern wireless sensors, thus replicating real-world conditions of functional movement.

⇒ While complete blinding is not possible post intervention, due to the nature of exercise, care will be taken to minimise investigator bias by using automated recording protocols.

## INTRODUCTION

Chronic musculoskeletal pain is a highly prevalent and complex condition that continues to challenge clinicians and academics and as such constitutes a major health and well-being burden.[1 2] The International Association for the Study of Pain defines chronic pain as 'pain which has persisted beyond normal tissue healing time', which, in the absence of other factors, is a pain condition lasting for more than 3 months.[3] A condition with chronic musculoskeletal pain may be referred to as chronic primary pain[4] and should be seen as a disease in its own right.[5] Chronic musculoskeletal pain can negatively affect physical movement and functioning,

sleep, social interaction, quality of life and work ability,[6] in essence all domains of life.

In the light of an active inference framework,[7–9] based on the free energy principle proposed by Friston[10 11] and resting on the concept of allostasis,[12] any living system becomes a model of the world it inhabits. Then, through the process of active inference, it attempts to maximise the proof of its own existence (chances of survival) by minimising prediction errors (variational free energy) through perception and action. The physiological concept of allostasis can be described as achieving stability through change,[12] in which the rate of change influences the process of adaption. In addition, living things may be defined by their ability to move autonomously, be it of bodies, limbs or cells, making movement a key driver of allostasis through sensory palpation of the world. In view of this, human movement, from cell respiration to locomotion, may constitute the major source of active inference on which the organism builds and maintains its model, making human movement of all sorts pertinent to long-term health and prosperity by facilitating the changes that are needed for stability.

Building on this notion, in a healthy context acute pain can be viewed as a precision signal for learning and control, with the main function being motivational to direct both short-term and long-term motor behaviour away from harm.[13] Chronic musculoskeletal pain, on the other hand, is suggestive of a living system in distress and may be explained by the creation and maintenance of abnormal priors that are afforded abnormal attention in a self-evidencing system.[7 14]

This reasoning is—we believe—in line with earlier theories such as the body-self neuromatrix model,[15] the theory of moving differently in pain[16] and effectively aligning with concepts such as sensitisation,[17 18] and ultimately depression[19] within chronic pain populations, as well as recent theory on active interoceptive inference.[20] Then chronic pain can also be viewed as the consequence of an allostatic overload from unresolved prediction errors of the system's lower level predictions about its internal and external milieu.[17 19 21–24] These prediction errors in lower level predictions may manifest themselves at higher levels as fear avoidance behaviour of movement[25] and intolerance of uncertainty,[26] which lead to the further reduction of physical activity.[27–29] This relative sensory deprivation[30 31] that is induced by sedentary behaviour may disable self-organisation normally driven through allostasis, leading to aberrant neurosignatures (models) of painful states. In essence, chronic musculoskeletal pain becomes a pathophysiology with a self-fulfilling hypothesis of pain as the systems only solution to minimise prediction errors. Building on this notion, physical activity and exercise, or simply human movement, can be viewed as active inference and as such may facilitate a return to allostasis and less painful future states by promoting the flexible reassignment of precision weighting that alters the individual's prediction about their body and their world.[32] This implies that adding physical activity or exercise to a rehabilitation programme may simply be reintroducing something essential that was engineered away generations ago.

From this active inference view of chronic pain, it is understandable that a biopsychosocial approach has gained traction in the treatment of chronic pain conditions,[6 33–35] since such interdisciplinary treatment delivered by a team of allied health professionals involves a combination of physical activity and exercise, psychological interventions, pharmaceutical treatments and patient education all acting as active inference[33] enabling a return to less painful states.

Given the fact that the musculoskeletal system is crucial to human locomotion or functional movement, a large body of research has focused on the effects of chronic pain on functional movement. There is now moderate-to-strong evidence that individuals with persistent low back pain demonstrate differences in gait kinematics,[34] that individuals with non-specific low back pain demonstrate impaired postural control when standing[35] and that individuals diagnosed with fibromyalgia demonstrate an altered gait compared with healthy individuals.[36] Neuroimaging studies have reported functional brain changes related to sensorimotor impairment in individuals with low back pain,[37] irritable bowel syndrome,[14] fibromyalgia[38] and migraine,[39] suggestive of shared disturbances of the motor system across different diagnoses. The overall finding is that persons with chronic pain conditions walk slower, with a lower step frequency, shorter step length and with a greater amplitude of back muscle activity.[34–36]

However, based on the notion of a shared disturbance, it is currently not known if the altered gait patterns identified in specific subgroups of chronic pain patients may generalise to wider populations of chronic musculoskeletal pain patients and if they could be used as markers of functional movement impairment in these populations. There is a paucity in how exercise, viewed as active inference, can have an acute effect on the motor control of walking in humans, especially in adaptions to perturbations such as faster walking speeds or dual tasks and if this is related to long-term sedentary behaviour.

Therefore, the main aim of this study is to investigate the acute effects of exercise as active inference on gait kinematics and activity of the back and neck muscles during diverse walking conditions in patients with chronic musculoskeletal pain compared with a reference sample consisting of participants without chronic musculoskeletal pain.

Our hypothesis is that a wide sample with chronic musculoskeletal pain will walk at lower speeds with lower step frequency and stride lengths but with increased muscle activity in the back and neck and that these differences will increase with task difficulty and begin to attenuate acutely by exercise.

## MATERIALS AND METHODS

### Trial design

The study consists of a two-sample two-armed parallel randomised controlled laboratory trial (RCT) that will be conducted at Dalarna University, Sweden (figure 1). The main sample consists of patients with chronic pain who will be randomised into one of two arms: an intervention group performing a single session of acute aerobic exercise (30 min stationary cycling) or a resting control group. The reference sample consists of participants without chronic musculoskeletal pain who will follow a similar RCT procedure, with the purpose being to investigate whether the acute response to exercise is unique for patients with chronic pain as our hypothesis proposes, or if it is just a general effect that would be seen in any population. The trial will comply with Consolidated Standards of Reporting Trials and Standard Protocol Items: Recommendations for Interventional Trials guidelines with Patient Reported Outcomes (SPIRIT).[40–42] A completed SPIRIT checklist for the trial can be found in online supplemental appendix 1. Data will be collected between January 2019 and May 2021.

For our primary outcome, a total estimated sample size of n=80 (n=20 in each of the four randomised groups) was determined based on previous studies using similar design and wearable technology.[43 44] A supplementary power calculation was conducted using normalised right-side erector spinae longissimus back muscle activity collected from the initial 20 participants to confirm the adequacy of the sample size. The analysis confirmed that a total sample size of n=80 is required to achieve a statistical power of 80% at a significance level of α=0.05 (two tailed) to detect a 25% difference between patients and participants without musculoskeletal pain.

### Participants and setting

This trial includes 80 participants. Patients (n=40) that visit primary and specialist pain clinics in Region Dalarna for evaluation or treatment of chronic musculoskeletal pain (> 3 months), as well as an age-matched and gender-matched convenience sample of participants without musculoskeletal pain (n=40), will be simultaneously recruited. The physiotherapists treating or evaluating potential participants at the cooperating clinics will act as liaisons to inform patients of the study and those interested in participation will be asked to provide their contact information. Researchers will then contact all potential participants by telephone and email prior to the laboratory visit, to perform a preliminary screening of eligibility criteria before scheduling a laboratory visit.

### Eligibility criteria

Participants aged 18–67 years will be included. Patients will be in assessment and/or treatment of chronic (>3 months) musculoskeletal (neck and/or low back) pain or widespread pain in primary or specialist pain care in Region Dalarna. Exclusion criteria: (1) any coexisting medical condition that would restrict safely participating in the exercise intervention, (2) previous spinal surgery, (3) inability to perform unsupported indoor walking, (4) inadequate Swedish or cognitive ability to provide consent and understand information and instructions, (5) pregnant in second or third trimester or childbirth in the last 3 months, (6) allergies to adhesive tape used for sensor application, (7) major hearing impairment effecting performance of auditory 1-back test, (7) pain caused by malignancies or systemic diseases, or localised pain in lower extremities only.

### Randomisation, allocation concealment and blinding

Participants will be randomly assigned to either the intervention or control group at a 1:1 allocation ratio, using stratified block randomisation with a fixed block size of six within the chronic musculoskeletal pain and no chronic pain strata. Allocation information is kept in opaque sealed envelopes and will not be disclosed to participants or researchers until all baseline walk trials are completed. All data collection will be automated in preprogrammed sequences and participant instructions read verbatim to minimise any investigator bias influencing participants' behaviour. Participants will not be informed of the hypotheses of the study, and the researcher performing the data analyses will be blinded to group allocation.

### Interventions

The acute aerobic exercise intervention will consist of one 20 min cycling interval preceded by a 5 min warm-up and followed by a 5 min cool-down (total duration of 30 min) and will be performed on a stationary electromagnetically braked cycle ergometer (Schoberer Rad Messtechnik, Jülich, Germany). Self-selected exercise intensity will be guided to moderate intensity by using Borg Ratings of Perceived Exertion (RPE) Scale with a target RPE of 13–15 (somewhat hard/hard).[45] Borg RPE and self-rated pain on Visual Analogue Scale (VAS) (0–100 mm) will be recorded every 5 min during the exercise session. Power output, cadence and heart rate will be recorded at 2 Hz. Participants will be able to freely regulate cadence and resistance to maintain the target RPE of 13–15, by means of a 14-increment mechanical gear shifter located on the handlebars. The ergometer is equipped with an upright handlebar, flat pedals and a wide-padded saddle with an extra gel cover to minimise sitting discomfort.

Participants allocated to the control groups will rest for the equivalent period of 30 min in a self-selected comfortable resting position accompanied by staff.

### Outcome measures

The primary outcomes (detailed in table 1) are kinematic measures of human movement and muscle activity that will be measured while participants are walking using the following three gait conditions: (1) self-selected habitual gait speed, (2) brisk gait speed and (3) self-selected habitual gait speed with a concurrent cognitive task. Our primary kinematic outcomes are gait speed, step frequency, stride length, lumbar rotation and gait

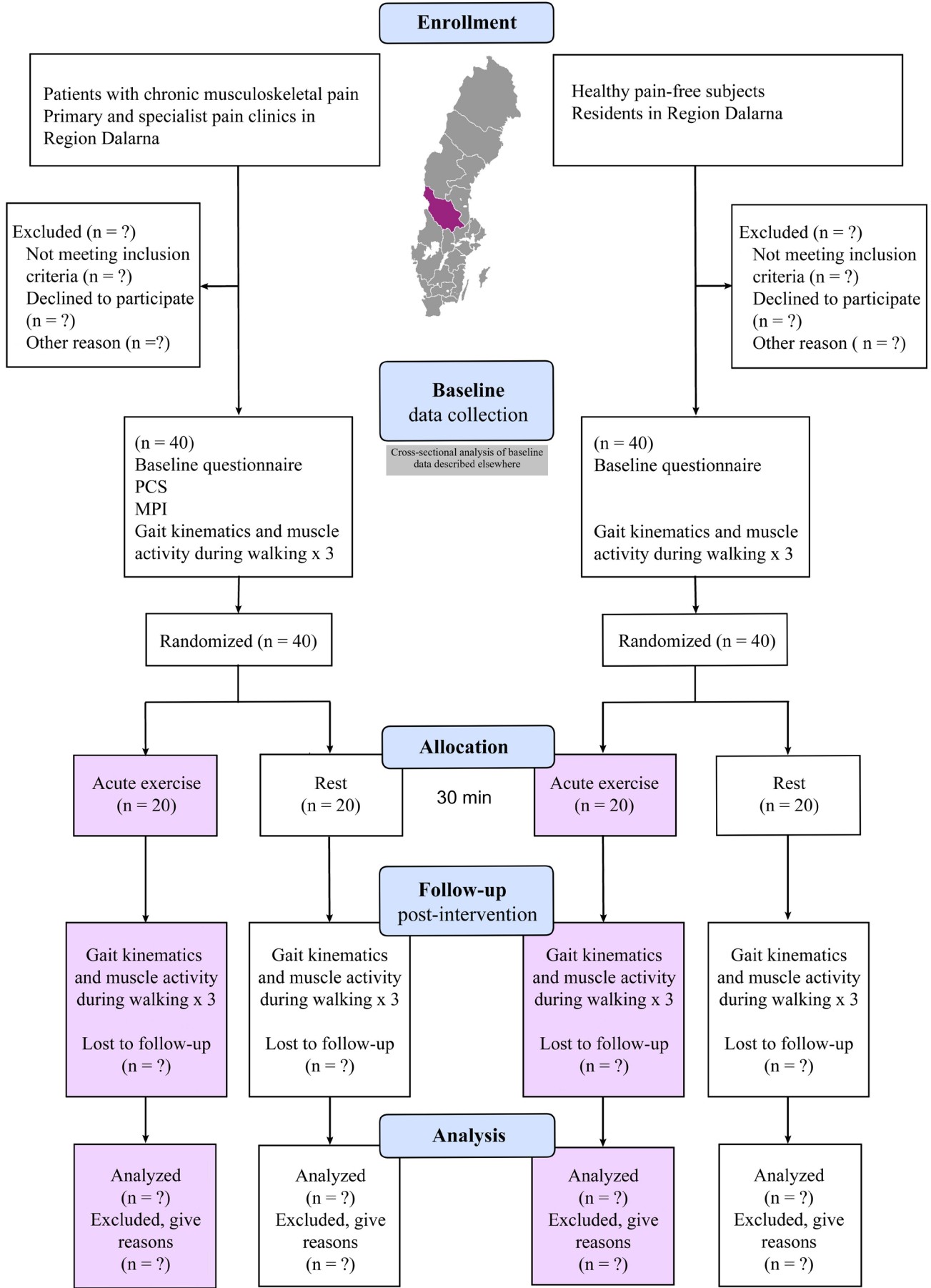

**Figure 1** Participant flowchart. MPI, Multidimensional Pain Inventory; PCS, Pain Catastrophizing Scale.

**Table 1** Outcomes and assessments

| | | | Baseline | Intervention | Follow-up |
|---|---|---|---|---|---|
| Baseline measures | Variable or assessment | Data Source | | | |
| Participant characteristics | Age, sex, body mass index (weight, height), educational level, work status, physical activity. | SAQ+PT | X | | |
| Pain catastrophising* | Pain Catastrophizing Scale (PCS) | SAQ | X | | |
| Effects on everyday life* | Multidimensional Pain Inventory (MPI) | SAQ | X | | |
| Primary outcome measures | | | | | |
| **Gait kinematics** ► Gait speed (m/s) ► Step frequency (steps/min) ► Stride length (cm) ► Lumbar rotation (°) ► Lateral step variability (three consecutive steps, the perpendicular deviation of the middle foot placement from the line connecting the first and the third step) **Back and neck muscle activity** ► Normalised mean muscle activity of all completed gait cycles per trial ► Back and neck normalised muscle activity at each 10 percentile of gait cycle | Gait kinematics with IMU-based sensors and wireless surface EMG, aggregated from minimum 50 gait cycles ► Self-selected habitual gait speed ► Brisk gait speed† ► Self-selected habitual gait speed with concurrent cognitive task (1-back test)† | PT | X | | X |
| Secondary outcome measures | | | | | |
| **Variability of gait kinematics** ► Gait speed (m/s) ► Step frequency (steps/min) ► Stride length (cm) ► Lumbar rotation (°) ► Lateral step variability **Variability of muscle activity** ► Normalised mean per trial ► Balance of left/right at each 10 percentile of gait cycle ► Back and neck normalised during stance and swing phase ► Number of muscle activity spikes (>100% normalised per complete gait cycle) | Non-linear analysis of gait kinematics with IMU-based sensors and wireless surface EMG, aggregated from minimum 50 gait cycles ► Self-selected habitual gait speed ► Brisk gait speed† ► Self-selected habitual gait speed with concurrent cognitive task (1-back test)† | PT | X | | X |
| Pain intensity* | VAS (0–100) | SAQ | X | X | X |

Data source: self-administered questionnaire (SAQ), physical testing (PT).
*Only measured in patients with chronic musculoskeletal pain.
†Brisk gait speed and self-selected gait speed with the 1-back test will be performed in random order.
EMG, electromyography; IMU, inertial monitoring unit; RPE, rating of perceived exertion (every 5 min during the intervention); VAS, Visual Analogue Scale (administered after every gait test).

stability, while primary outcomes for the muscle activity domain will be mean normalised electromyography (EMG) amplitude averaged over aggregated gait cycles.

Our secondary outcome measures are variability of gait kinematics and muscle activity and self-rated current pain intensity using a horizontal 100 mm line VAS with the statement 'no pain at all' at the extreme left and 'the worst possible pain' at the extreme right (chronic pain sample only).[46 47]

## Trial procedure, instruments and data collection

The entire trial procedure at the laboratory is expected to take approximately 3 hours and will be supervised by two researchers. On arrival at the laboratory, eligibility

screening, presentation of the study procedures and collection of informed consent (online supplemental appendix 2) will be performed. Participants will complete the following questionnaires before commencing any testing: (1) participant characteristics (age, sex, educational level, occupational status, pain locations, pain duration) including two validated items with questions about physical activity.[48] Participants with chronic musculoskeletal pain will also complete the following two questionnaires relating to pain experiences: (2) the Pain Catastrophizing Scale,[49] which contains 13 items regarding participants' thoughts and feelings associated with pain, with a total score ranging from 0 to 52, with higher scores representing higher pain catastrophising and (3) the West Haven-Yale Multidimensional Pain Inventory, which is a questionnaire that measures the psychosocial, cognitive and behavioural effects of chronic pain.[50–53]

A researcher will then record participant body height using a stadiometer (Harpenden Stadiometer, Holtain Limited, Crosswell, UK) and body weight using a weighing scale (Midrics, Sartorius AG, Goettingen, Germany). Participants will be introduced to the 1-back test[54] and listen to a 30 s practice sequence, then two 30 s 1-back baseline tests will be recorded with participants in a seated position. The 1-back test will be performed with a pair of minimally intrusive wireless handsfree on-ear headphones connected via Bluetooth to a PC using Audacity Software V.2.3.0. Participants will listen to an audio file consisting of a randomised number sequence and will be instructed to respond as quickly and accurately as possible to audio prompts, repeating the digit before the last one in the consecutive sequence during the entire 1-back walking trial. All responses will be recorded.

After this, seven wireless inertial monitoring unit sensors (OPAL, APDM, Oregon, USA) incorporating a three-axis gyroscope, accelerometer and magnetometer will be applied on participants' limbs using elastic straps as specified in figure 2. Gait kinematics will be streamed and logged wirelessly from each sensor at 128 Hz during all trials and will be automatically processed via Moveo Explorer software to extract gait variables specified as primary or secondary outcomes according to table 1.[55] Participants will be given standardised instructions prior to starting each gait trial in order to comply with the initial calibration procedure described by the manufacturer.

Wireless EMG sensors (Delsys Trigno, Delsys, Boston, Massachusetts, USA) will be positioned according to anatomical landmarks detailed in figure 3, as per surface EMG for the non-invasive assessment of muscles guidelines[56 57] and data will be recorded at 1111 Hz. Skin will be shaved and cleansed with a chlorhexidine digluconate ethanol cutaneous solution (Klorhexidinsprit 5 mg/mL, Fresenius Kabi AB, Uppsala, Sweden). To obtain normalised (reference) muscle activity, each participant will perform four standardised submaximal standing trunk flexion (hip flexion to 45° from neutral standing position) muscle contraction tests, interspersed by a 60 s rest. EMG sensors will be removed after baseline measurements and

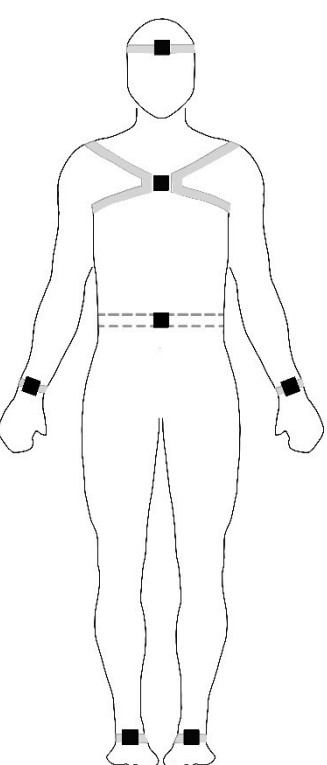

**Figure 2** Inertial monitoring unit sensor placement on the forehead, sternum, lumbar spine, wrists and feet. Adapted with permission by APDM.

reapplied prior to the follow-up walking trials. Before removal, sensor positions will be marked with a marker pen to facilitate accurate reapplication.

All gait trials will be performed indoors in a continuous fashion on a 15 m standardised walkway with tape markings on the floor indicating starting and 180° turning points. Trials will have a minimum duration of 2 min and finish when participants cross the start line with a time exceeding 2 min. Participants will initially perform a familiarisation trial at a self-selected habitual speed followed by a short rest if needed, to avoid any accumulated pain or fatigue. After this, participants will proceed to walk the 15 m walkway under the three different conditions described above: (1) an initial self-selected habitual gait speed, (2) a brisk gait speed and (3) a self-selected habitual gait speed with a concurrent cognitive task (1-back test). The order of performance of the brisk and 1-back walking conditions (2 or 3) will be randomised and the allocation will be kept concealed from the participants until the initial condition with self-selected habitual gait speed has been performed. For the brisk walking condition, participants will be instructed to walk faster than they habitually do, as they imagine themselves doing when slightly late for a bus they want to catch. The self-rated pain intensity (VAS) of participants with chronic musculoskeletal pain will be recorded after each walk trial. The risk of complications due to measurements or interventions is considered to be low, but any adverse or serious adverse events will be logged and reported as well as covered by laboratory insurance.

| Muscle | EMG electrode locations | Orientation | Illustration |
|---|---|---|---|
| Erector Spinae Neck, bilateral | Placed 20 mm lateral to the median line at the level of C4-C5 spinous process | Vertical | |
| Erector Spinae Longissimus (EL), bilateral | Placed 2 finger widths lateral from the spinous process of L1. | Vertical | |
| Erector Spinae Iliocostalis (IL), bilateral | Placed 1 finger width medial from a line from the posterior spina iliaca superior to the lowest point of the lower rib, at the level of L2. | Vertical | |
| Multifidus (MF), bilateral | Placed on and aligned with a line from caudal tip posterior spina iliaca superior to the interspace between L1 and L2 interspace at the level of L5 spinous process (i.e. about 20-30 mm from midline). | Aligned with line, 45° from horizontal | |

EMG, electromyography

**Figure 3** Overview wireless electrode placement for surface EMG recording of the trunk and neck musculature according to surface electromyography for the non-invasive assessment of muscles guidelines (http://www.seniam.org/).

## Data management

Participant questionnaires will be checked for completeness and congruity before data is entered into the database. Data will be stored on secure university servers and in locked cabinets only available to the research team. Gait kinematics will be aggregated for all valid gait cycles of each gait condition. For each individual, a minimum of 50 gait cycles will be used and aggregated prior to analysis.[58] Back and neck muscle activity (root mean square (RMS)) synchronised to gait cycles and aggregated from each gait condition will be normalised as percentage RMS against reference contractions.

## Statistical analysis

Participant characteristics will be reported using descriptive statistics. The statistical analyses will be performed on an intention-to-treat basis and potential differences between the intervention groups and the control groups will be analysed. Analysis will primarily be conducted with variance analysis with respect to both data quality and distribution (data assumptions) and with the addition of repeated measurements where applicable. Log transformations of data will be made as appropriate. Data will be presented with means and variances. To explore the magnitude of the effect of interventions, effect sizes will be calculated.[59] Lastly, non-linear analysis using Gait Evaluation Differential Entropy Method[60] will be used to assess variability of gait kinematic and muscular activity.[60–62]

## Ethics and dissemination

The study has been approved by the Regional Ethics Review Board in Uppsala, Sweden (#2018/307). Findings will be disseminated via conference presentations, publications in peer-reviewed journals and engagement with patient support groups and clinicians.

## Patient and public involvement

The burden of the study procedures has been assessed by patients during preliminary pilot work, primarily to assess the following areas: the feasibility of the cycling exercise intervention, the baseline questionnaires and the instrumentation to capture gait kinematics and EMG data. Study participants will be invited to a briefing where the results of the study will be presented. In addition, study participants will be given a summary of the results written in Swedish. Results will be presented at the annual meeting of the Swedish National Registry for Chronic Pain Rehabilitation where patient groups are represented.

## DISCUSSION

To our knowledge, the current study is the first RCT to investigate the acute effect of aerobic exercise on gait kinematics and muscle activity in participants with chronic musculoskeletal pain compared with reference participants from a perspective of exercise as active inference. Our two-sample two-armed parallel design allows

for a cross comparison of effects in patients with chronic musculoskeletal pain and healthy participants that will follow the same study protocol and will contribute to our understanding of how chronic musculoskeletal pain manifests on measures of human motor behaviour and how exercise can influence this in a rehabilitation setting. Knowledge of how gait kinematics and back and neck muscular activity may change during different gait conditions and from aerobic exercise may contribute to the understanding of how exercise therapy influences our generative models in the process attenuates allostatic load through active inference. The use of modern, miniaturised and completely wireless sensors during overground walking replicates real-world conditions, hence strengthening the ecological validity of our results. Objective measurements of movement and muscular activity during diverse walking conditions have been chosen as primary outcomes since such a methodology can both uniquely and sensitively show and quantify deviating activation patterns as validly reflected in everyday functional movements and commonly performed physical activities.

Selection bias is difficult to completely overcome in exercise intervention studies as volunteering participants will likely be positively biased towards exercise in general and motivation to participate in an exercise intervention is also likely influenced by the severity of symptoms experienced when asked to volunteer for the study. We will compare and validate our sample baseline characteristics with a new nationwide large database including 60 000 patients with chronic pain.[63]

Concerning our reference participants without chronic musculoskeletal pain, we aim to create a sample that is sex and age matched with our chronic musculoskeletal pain sample and will serve as a reference sample for all our outcomes.

Viewing this exercise intervention as active inference will hopefully add knowledge on how acute movement through exercise may promote an individual's predictions about their body and their world by creating a platform for belief updating. Our use of a cycling modality with a rather different muscle recruitment and joint loading pattern compared with walking may be beneficial, especially since stationary cycling may be more palatable to participants than running at the same intensity as it may lead to an undesired level of fatigue, fear or increased pain.

In addition, the aerobic exercise intervention in the current study reflects recommendations in exercise guidelines and allows an easy progression of rehabilitation in terms of both intensity and duration.[64] Since a bicycle can be used for therapy, recreation and transportation, there are opportunities for healthcare professionals to facilitate changes in lifestyle that have the potential to mitigate morbidity and function in patients with chronic pain,[65] simultaneously reducing the risks and impact of common comorbidities.

With the foundation of this project being objective measures of overground walking, it adheres well to the Initiative on Methods, Measurement and Pain Assessment in Clinical Trials, which recommends that all chronic pain trials should include measures of physical functioning.[47] This project is expected to result in increased knowledge of how acute exercise affects subsequent functional tasks, here measured during different walking conditions.

Our proposed project will deepen our understanding on the topic of alterations in functional movement in chronic musculoskeletal pain conditions. Results from the current study may help us reach a better understanding of how acute aerobic exercise acts as active inference to attenuate chronic pain by facilitating allostasis. This knowledge will help us formulate better questions as to how generative models in chronic pain conditions are formed, maintained and updated through movement, or lack thereof.

**Author affiliations**
¹School of Health and Welfare, Dalarna University, Falun, Sweden
²School of Information and Engineering, Dalarna University, Borlänge, Sweden
³Department of Neurobiology, Care Sciences and Society, Division of Physiotherapy, Karolinska Institutet, Huddinge, Sweden
⁴Medical unit Occupational therapy & Physiotherapy, Theme Women's Health and Allied Health Professional, Karolinska University Hospital, Stockholm, Sweden
⁵Center for Clinical Research Dalarna, Uppsala University, Region Dalarna, Falun, Sweden
⁶Department of Health, Medicine and Caring Sciences, Pain and Rehabilitation Centre, Linköping University, Linköping, Sweden
⁷Regional Board Administration, Region Dalarna, Falun, Sweden

**Contributors** JW, VS, BOÄ and LV took responsibility for the integrity of the study. All authors were involved in the study's concept and design. JW and VS were involved in acquisition of data. JW, PE, AM, RL, DMC, RGN and BOÄ were involved in drafting and revising the article critically for important intellectual content. All authors were involved in manuscript revision and read and approved the final version to be published.

**Funding** This work is supported by the Swedish Research Council for Health (2015-02512), Working Life and Welfare (FORTE; 2017-00177) and Dalarna University (N/A), Falun, Sweden. The funders had no role in study design and will have no role in any part of the study or reporting of the results.

**Competing interests** None declared.

**Patient and public involvement** Patients and/or the public were involved in the design, or conduct, or reporting, or dissemination plans of this research. Refer to the Materials and methods section for further details.

**Patient consent for publication** Not applicable.

**Ethics approval** The study has been approved by the Regional Ethics Review Board in Uppsala, Sweden (#2018/307).

**Provenance and peer review** Not commissioned; externally peer reviewed.

**ORCID iDs**

Jens Westergren http://orcid.org/0000-0001-8356-4669
Veronica Sjöberg http://orcid.org/0000-0002-3843-0407
Linda Vixner http://orcid.org/0000-0002-6923-7140
Roger G Nyberg http://orcid.org/0000-0003-4812-4988
David Moulaee Conradsson http://orcid.org/0000-0003-0003-5513
Andreas Monnier http://orcid.org/0000-0001-8923-1670
Riccardo LoMartire http://orcid.org/0000-0003-0672-8214
Paul Enthoven http://orcid.org/0000-0003-3707-5869
Björn O Äng http://orcid.org/0000-0001-7767-4589

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
