## [Reviewer comments · BMJ Open]

ARTICLE DETAILS

TITLE (PROVISIONAL)	Acute exercise as active inference in chronic musculoskeletal pain, effects on gait kinematics and muscular activity in patients and healthy participants – A study protocol for a randomised controlled laboratory trial
AUTHORS	Westergren, Jens; Sjöberg, Veronica; Vixner, Linda; Nyberg, Roger; Conradsson, David; Monnier, Andreas; LoMartire, Riccardo; Enthoven, Paul; Ång, Björn

VERSION 1 – REVIEW

REVIEWER	Sas, Senem Erciyes Universitesi
REVIEW RETURNED	26-Dec-2022

GENERAL COMMENTS	The manuscript presents the acute effects of physical activity on gait kinematics and muscle activity of the lumbar and cervical region in patients with chronic pain. It can be accepted in its present form.
---

REVIEWER	Ribeiro, Ana Universidade de São Paulo Faculdade de Medicina, Physical Therapy, Speech and Occupational Therapy Department
REVIEW RETURNED	29-Dec-2022

GENERAL COMMENTS	General comments on the paper: Scientific article is very important for chronic musculoskeletal pain area. The authors bring important contributions to comprehension acute effects of exercise as active inference on gait kinematics and activity of the back and neck muscles during diverse walking conditions in patients with chronic musculoskeletal pain. The introduction is well described with a clear line of reasoning and evidence from the literature, but I suggest adding the issue of chronic pain to low back pain and not just fibromyalgia. The effects and benefits of aerobic exercises need to be detailed, especially for biomechanical variables. The study hypothesis needs to be rewritten, it is very confusing, for example what the authors mean by heterogeneous sample, increases in head and neck muscle activity and the issue of exercise, the introduction does not raise all questions of the authors' hypotheses. This needs to be reviewed. The methodology needs to be completely rewritten adding information that is relevant for a better understanding of what was
---

	done in this study. The authors do not make clear the diagnosis of chronic musculoskeletal pain in the back and neck and what was the pain intensity to participate in the study, since this information may influence the measured variables. These issues need to be clarified. Better establish sample allocation and blinding. Gait assessment and the instruments used need to be better described. Specify which muscles will be evaluated for Back and neck muscle activity. In the discussion, the session of participants needs to be better detailed in the profile of patients with chronic musculoskeletal pain (disease and pain intensity) and the control as well. The intervention is very well contextualized. The potential of the study can be improved. In the variables, it is necessary to clarify in detail the effect of the type of acute aerobic exercise on each variable to be analyzed. The conclusion session, the authors need rewrite based on the specifics of the proposed objectives and analyzed variables.
--	--

VERSION 1 – AUTHOR RESPONSE

Reviewer: 1

Dr. Senem Sas, Erciyes Universitesi

Comments to the Author:

The manuscript presents the acute effects of physical activity on gait kinematics and muscle activity of the lumbar and cervical region in patients with chronic pain.

It can be accepted in its present form.

Author response and action:

Thank you for the comment. Our hope is that this protocol can spark an interest in active inference as a bridge between lines of research within the field of chronic pain.

Reviewer: 2

Dr. Ana Ribeiro, Universidade de São Paulo Faculdade de Medicina, Universidade de Santo Amaro Faculdade de Medicina

Comments to the Author:

General comments on the paper:

Scientific article is very important for chronic musculoskeletal pain area. The authors bring important contributions to comprehension acute effects of exercise as active inference on gait kinematics and activity of the back and neck muscles during diverse walking conditions in patients with chronic musculoskeletal pain.

Author response and action: Thank you, we hope that this protocol and following papers can introduce a refreshing view of the complex relationship between chronic musculoskeletal pain and

human movement and at the same time anchor the construct of active inference in contemporary research and practice within the field of musculoskeletal pain.

The introduction is well described with a clear line of reasoning and evidence from the literature, but I suggest adding the issue of chronic pain to low back pain and not just fibromyalgia. The effects and benefits of aerobic exercises need to be detailed, especially for biomechanical variables. The study hypothesis needs to be rewritten, it is very confusing, for example what the authors mean by heterogeneous sample, increases in head and neck muscle activity and the issue of exercise, the introduction does not raise all questions of the authors' hypotheses. This needs to be reviewed.

Author response and action:

We agree that ending the paragraph with a fibromyalgia reference placed unnecessary focus on this condition, since the rest of the paragraph emphasized low back pain. We have now edited this in line with reviewer suggestion by ending the paragraph in a more neutral way, also referencing by referencing two sources. What we really want to convey in the introduction is our belief that when pain has become chronic and usually widespread, initial anatomical pain location is of lesser importance, thus motivating our choice to include a natural range of people seeking health care for chronic musculoskeletal pain. Hence, our participants were not recruited based on specific diagnoses, nor by pain intensity, we will rather learn of their pain intensity and pain locations upon analysis of our data.

The methodology needs to be completely rewritten adding information that is relevant for a better understanding of what was done in this study. The authors do not make clear the diagnosis of chronic musculoskeletal pain in the back and neck and what was the pain intensity to participate in the study, since this information may influence the measured variables. These issues need to be clarified. Better establish sample allocation and blinding. Gait assessment and the instruments used need to be better described. Specify which muscles will be evaluated for Back and neck muscle activity.

Author response and action:

Thank you for highlighting this. We hope our reply above has clarified the question regarding diagnosis and pain intensity upon recruitment. We believe this strengthens the validity of our findings since our recruited sample is based on real world health care seekers suffering from chronic musculoskeletal pain. We have also clarified our methods regarding trial design on page 9 and sample allocation and blinding on page 10. In addition, on page 16 we have improved our description of the automated gait assessment and added a reference paper describing our IMU instrumentation further.

With regards to the specification of surface EMG locations we realize that our previously submitted Figure 3 could have received a stronger mention in the running text, so we have edited this to attract the reader's attention to it. In Figure 3, we specify the exact method for electrode placement written as per SENIAM guidelines for each of the four bilaterally measured muscles, including a picture illustration.

In the discussion, the session of participants needs to be better detailed in the profile of patients with chronic musculoskeletal pain (disease and pain intensity) and the control as well. The intervention is

very well contextualized. The potential of the study can be improved. In the variables, it is necessary to clarify in detail the effect of the type of acute aerobic exercise on each variable to be analyzed.

Author response and action:

Thank you, we agree that this needed some clarification. In accordance with our previous changes above regarding diagnosis and pain intensity we have edited the corresponding participant section in our discussion to reflect this.

The conclusion section, the authors need rewrite based on the specifics of the proposed objectives and analyzed variables.

Author response and action:

The conclusion section has been removed upon advice from the editor. Thank you.

Reviewer: 1

Competing interests of Reviewer: No

Reviewer: 2

Competing interests of Reviewer: - No applicable